# Diverse and abundant phages exploit conjugative plasmids

Natalia Quinones-Olvera [1,2,3,8], Siân V. Owen [1,2,3,8] ✉, Lucy M. McCully [1,2,3], Maximillian G. Marin[1], Eleanor A. Rand [1,2,3], Alice C. Fan [1,2,3,4,5], Oluremi J. Martins Dosumu[3,6], Kay Paul [3,6], Cleotilde E. Sanchez Castaño[3,6], Rachel Petherbridge [5], Jillian S. Paull[5,7] & Michael Baym [1,2,3,7] ✉

Phages exert profound evolutionary pressure on bacteria by interacting with receptors on the cell surface to initiate infection. While the majority of phages use chromosomally encoded cell surface structures as receptors, plasmid-dependent phages exploit plasmid-encoded conjugation proteins, making their host range dependent on horizontal transfer of the plasmid. Despite their unique biology and biotechnological significance, only a small number of plasmid-dependent phages have been characterized. Here we systematically search for new plasmid-dependent phages targeting IncP and IncF plasmids using a targeted discovery platform, and find that they are common and abundant in wastewater, and largely unexplored in terms of their genetic diversity. Plasmid-dependent phages are enriched in non-canonical types of phages, and all but one of the 65 phages we isolated were non-tailed, and members of the lipid-containing tectiviruses, ssDNA filamentous phages or ssRNA phages. We show that plasmid-dependent tectiviruses exhibit profound differences in their host range which is associated with variation in the phage holin protein. Despite their relatively high abundance in wastewater, plasmid-dependent tectiviruses are missed by metaviromic analyses, underscoring the continued importance of culture-based phage discovery. Finally, we identify a tailed phage dependent on the IncF plasmid, and find related structural genes in phages that use the orthogonal type 4 pilus as a receptor, highlighting the evolutionarily promiscuous use of these distinct contractile structures by multiple groups of phages. Taken together, these results indicate plasmid-dependent phages play an under-appreciated evolutionary role in constraining horizontal gene transfer via conjugative plasmids.

Viral infections pose a constant threat to the majority of life on Earth[1,2]. Viruses recognize their hosts by interacting with structures (receptors) on the cell surface[3]. For viruses that infect bacteria (phages), these receptors are usually encoded on the chromosome, and are part of core cellular processes including transport proteins or structurally integral lipopolysaccharides[4]. However, certain mobile genetic elements, such as conjugative plasmids, also contribute to the cell surface landscape by building secretory structures (e.g., type 4 secretion systems) which enable transfer into neighboring bacterial cells[5,6]. Plasmid-dependent phages (PDPs) have evolved to use these plasmid-encoded structures as receptors, and can only infect plasmid-containing bacteria[7]. Conjugative plasmids can often transmit

between distantly related bacterial cells, creating new phage-susceptible hosts by horizontal transfer of receptors[8].

Almost all previously identified PDPs belong to unusual "non-tailed" groups of phages, some of which have more in common with eukaryotic viruses than the "tailed" phages that make up the majority of bacterial virus collections[9,10]. This includes the dsDNA alphatectiviruses, and members of the ssDNA inoviruses and ssRNA fiersviruses. The handful of known PDPs have had profound impacts on the field of molecular biology, enabling phage display technology[11] (F plasmid-dependent phage M13), and in vivo RNA imaging[12] (F plasmid-dependent phage MS2). PDPs have also aided in our understanding of the origin of viruses: tectiviruses are thought to represent ancient ancestors of adenoviruses[13].

Predation by PDPs exerts strong selection on bacteria to lose conjugative plasmids, or to mutate/repress the conjugation machinery including the pilus[14–17]. As antibiotic resistance genes are frequently carried and spread by conjugative plasmids[18–21], selection against plasmid carriage functionally selects against antibiotic resistance in many instances. The extent to which this is a significant evolutionary pressure on antibiotic resistance depends on how frequent these phages are in nature.

Despite the remarkable properties of these phages and their intriguing association with conjugative plasmids, only a handful of PDPs exist in culture. In the 1970s–80s at least 39 different PDPs were reported targeting 17 different plasmid types (classified by "incompatibility" groups)[7]. However, most of these reports predated the era of genome sequencing, and to our knowledge, most of the reported PDPs have been lost to science. Here, we use a targeted discovery approach to show that PDPs are easily discoverable in the environment, and associated with unappreciated genetic and phenotypic diversity.

## Results

### Co-culture enables direct discovery of plasmid-dependent phages

PDPs have historically been identified and quantified by taking phage collections that were isolated on bacteria containing conjugative plasmids and screening them on isogenic plasmid-free bacteria, to look for plasmid-specific phenotypes[22]. As PDPs use the proteins expressed by conjugative plasmids as receptors, their host range mirrors plasmid host range, and typically crosses bacterial genera[7]. Exploiting this property, some studies have used multi-species enrichment methods to increase the likelihood of finding PDPs that can infect multiple different plasmid-containing hosts[23,24]. Alternatively, relative enrichment of PDPs by the depletion of species-specific phages, so called "somatic" phages, has been described[25]. However, these enrichment methods do not allow the direct quantification of PDPs relative to somatic phages in a sample, and suffer from other drawbacks such as increased likelihood of repeated isolation of the same phage, and bias against PDPs that may use both species-specific and plasmid-encoded receptors. In order to isolate and directly assess the abundance and diversity of PDPs in the environment, we set out to develop a targeted isolation approach. The challenge of targeted isolation is discriminating PDPs, in a direct, non-labor-intensive way, from somatic phages, which may be more or less abundant than PDPs depending on the environmental sample, the plasmid in question and the host species[26].

To differentiate PDPs, we co-cultured *Salmonella enterica* and *Pseudomonas putida*, a pair of taxonomically distinct bacteria with no known shared phages, that grew well in coculture. We also selected a known PDP, the *Alphatectivirus* PRD1, which depends on IncP group conjugative plasmids such as RP4 and pKJK5, and can infect *S. enterica* and *P. putida* provided they contain an IncP plasmid. We made a modification to the traditional phage plaque assay, by co-culturing these strains with differential fluorescent tags, together in the same soft-agar lawn. After applying dilutions of phages, the plaquing phenotype of the PDP PRD1, which efficiently killed both fluorescently labeled strains in the lawn (resulting in no fluorescent signal) was immediately discernible from species-specific phage 9NA (infecting *S. enterica*) and SVOΦ44 (infecting *P. putida*) (Fig. 1a). This observation formed the basis of the targeted phage discovery method we termed "Phage discovery by coculture" (Phage DisCo) (Fig. 1b).

To directly isolate PDPs dependent on the IncP plasmids using Phage DisCo, environmental samples putatively containing PDPs can be mixed together with fluorescently labeled *S. enterica* and *P. putida* strains containing the IncP conjugative plasmid RP4 (Fig. 1b). After growth of the bacterial lawn, phages are immediately identifiable by the fluorescence phenotype of their plaques: *P. putida* phages appear as red plaques where only *S. enterica* RP4 (red) is able to grow, *S. enterica* phages appear as blue plaques where only *P. putida* RP4 (blue) is able to grow, and PDPs make dark plaques where both bacteria in the lawn are killed (Fig. 1b). As a proof of principle, we mixed equimolar amounts of the test phages, 9NA, SVOΦ44, and PRD1, to simulate an environmental sample containing both species-specific phages and PDPs (Fig. 1c). After incubation and growth of the bacteria in the lawn, the plate was photographed using a custom fluorescence imaging setup. Once the two fluorescent image channels were digitally merged, plaques of all three phages were easy to identify by fluorescence phenotype, and importantly, the PRD1 plaques could be easily discerned from the plaques made by the two species-specific phages.

Having established the efficacy of the phage DisCo method using phages we had in culture, we set out to look for new PDPs in samples collected from compost, farm waste, and wastewater in the Greater Boston area (MA, USA) (Supplementary Dataset 1e). We chose to focus on phages depending on conjugative plasmids of the IncP and IncF incompatibility groups. IncP and IncF plasmids are also associated with extensive antibiotic resistance gene cargo and are frequently isolated from environmental[27] or clinical[28,29] samples, respectively. The archetypal IncF plasmid, the F plasmid originally isolated from *E. coli* K12, has a narrower host range than IncP plasmids, so we changed the DisCo hosts strains to *E. coli* and *S. enterica*. As *S. enterica* strains natively encode an IncF plasmid, we used a derivative that had been cured of all plasmids and prophages to mitigate any interference from these elements. Initially we collected 50 novel unique phages dependent on the IncP plasmid, and 13 dependent on the IncF plasmid. In order to identify any narrow-host range phages that may have been missed in our IncP-search using *S. enterica* and *P. putida*, we capitalized on the low abundance of *P. putida* phage in wastewater to perform a traditional single-host phage screen that captured an additional two narrow-host range phages dependent on the IncP plasmid in *P. putida*. Therefore, in total, we collected 65 novel PDPs in this study (Fig. 1d). All phages were further characterized by genome sequencing and we adopted a naming system wherein each phage was given a unique color identifier with a prefix consistent with previously isolated PDPs.

### IncP plasmid-dependent tectiviruses from a limited geographic area fully encompass the previously known global diversity

Genomic analysis revealed that 51 of the 52 IncP-dependent phages in our collection belonged to the *Alphatectivirus* genus in the *Tectiviridae* family, and are related to Enterobacteriophage PRD1. Surprisingly, despite our sampling being limited to a small geographical area and short time frame, the phages we isolated represented significantly more genomic diversity than the six previously known plasmid-dependent tectiviruses that were isolated across multiple continents, suggesting these phages are greatly under sampled. We estimate our collection expands the genus *Alphatectivirus* from two species (represented by type isolates PRD1 and PR4) to 12, as determined by pairwise nucleotide identity of all alphatectiviruses, including the six previously known alphatectiviruses and our 51 new isolates (Fig. S1a) (species cutoff <95% nucleotide identity, according to guidelines

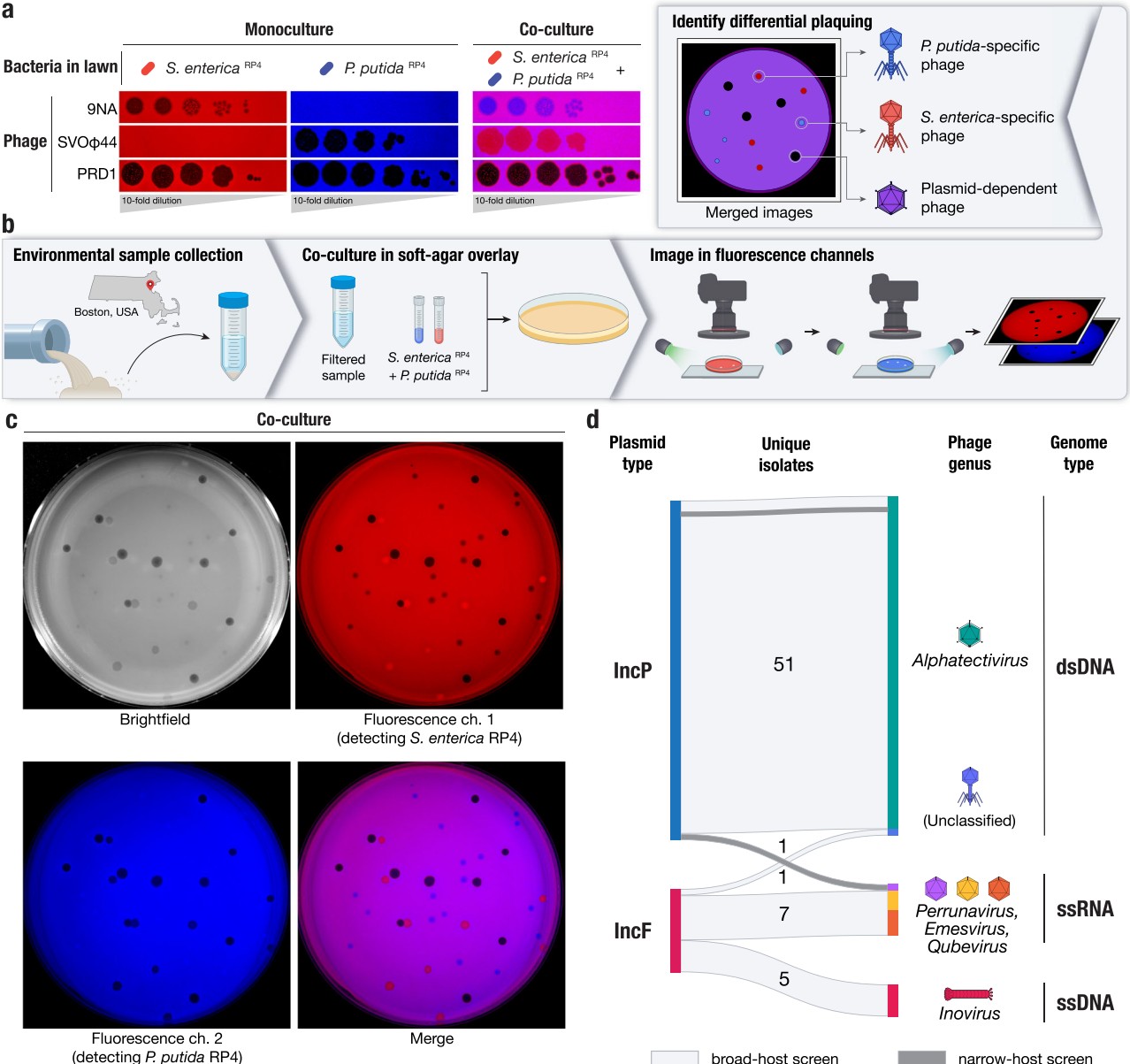

**Fig. 1 | A method for systematic discovery of plasmid-dependent phages by fluorescence assisted co-culture (Phage DisCo). a** Comparison between mono-cultured lawns and a co-cultured lawn. All images show merged GFP and mScarlet fluorescence channels (GFP shown in blue for visualization purposes). In mono-cultured lawns with exclusively *S. enterica* RP4 (red) or *P. putida* RP4 (blue), only plasmid-dependent phage PRD1 and the appropriate species-specific phages (*S. enterica* phage 9NA or *P. putida* phage SVOφ44) generate plaques. In the co-culture lawn (magenta, showing the overlap of both bacterial hosts), the species-specific phages form plaques on one of the species while plasmid-dependent phage PRD1 forms plaques on both species. **b** Schematic of the Phage DisCo method and screening strategy. Environmental samples were collected from around Boston, USA, and processed into a co-culture lawn with two plasmid-carrying bacterial hosts labeled with different fluorescent markers. After incubation, the plates were imaged in both fluorescence channels. The merged image was then used to distinguish species-specific phages (forming red or blue plaques) from plasmid-dependent phages (forming dark plaques). **c** Imaging of co-cultured lawn with white light or fluorescent light channels, with approximately equimolar concentrations of phages shown in (**a**) to simulate a screening plate from an environmental sample containing plasmid-dependent and species-specific phages. Individual plaques are clearly discernible as 9NA (blue plaques), SVOφ44 (red plaques), and PRD1 (dark plaques). **d** Sankey diagram summarizing the properties of the 65 novel plasmid-dependent phages isolated in this study, broken down by plasmid-dependency, phage genus, and genome type. In total, 63 phages were obtained from the broad-host range phage DisCo screen (light gray horizontal blocks), and an additional two phages were isolated from a narrow-host range screen (dark gray horizontal blocks).

published by the International Committee on Taxonomy of Viruses (ICTV)[30].

Additionally, by querying genome databases we identified one published tectivirus genome, *Burkholderia phage BCE1*, closely related to PRD1 by whole genome phylogeny (Fig. 2a). As *Burkholderia sp.* are known hosts of IncP-type conjugative plasmids[31] we expect that the *Burkholderia cenocepacia* host used to isolate BCE1 carried such a plasmid (highlighting the serendipitous nature by which PDPs are

often found) and we include BCE1 in our known plasmid-dependent tectivirus phylogeny. Novel conjugative plasmids have recently been detected in *Burkholderia contaminans* isolates, after their existence was implicated by the isolation of alphatectiviruses on these strains[32], suggesting *Burkholderia* species may be common hosts for these plasmids and phages.

While the new plasmid-dependent tectiviruses we report expand the known diversity of this group of phages from two to twelve

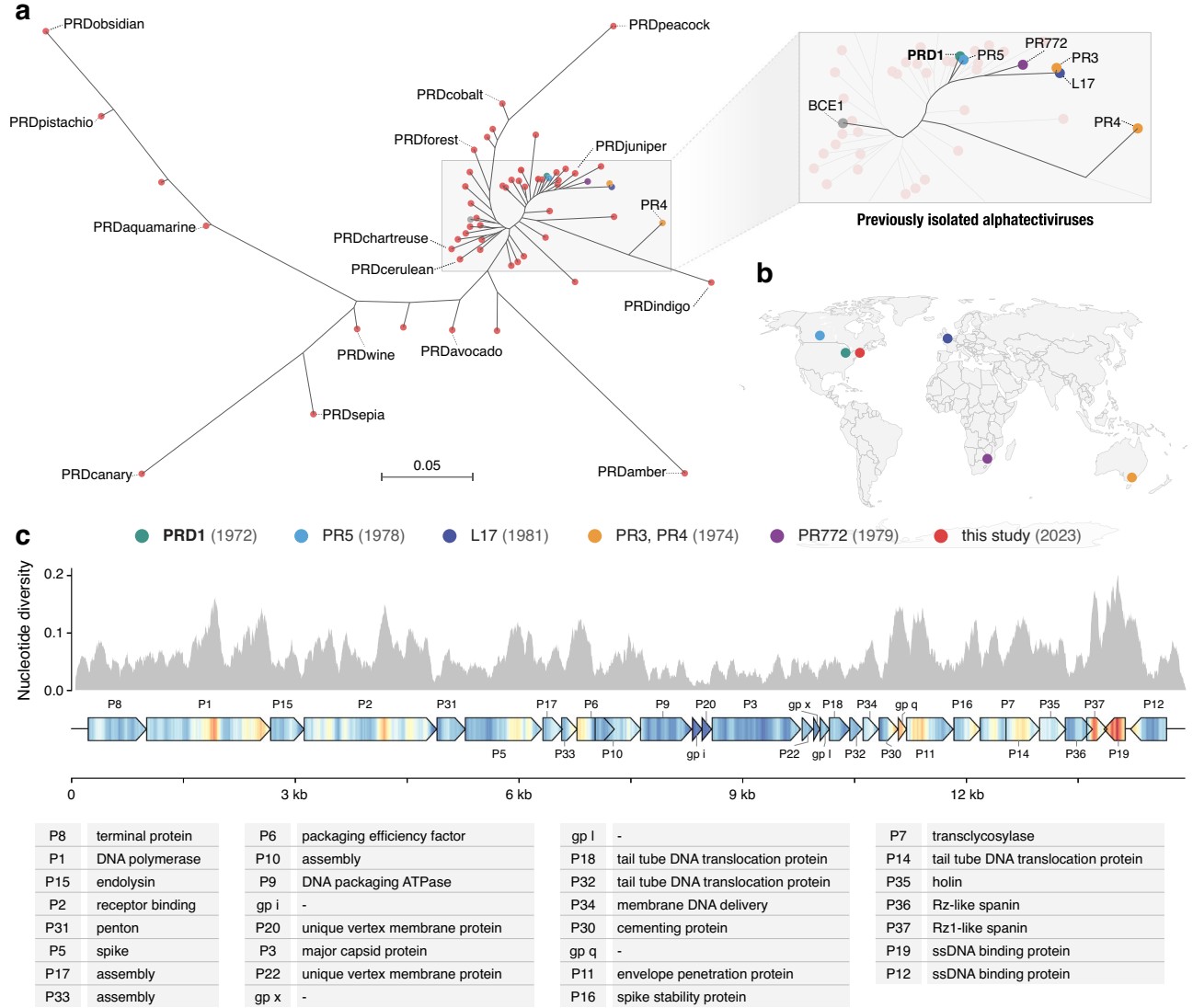

**Fig. 2 | Targeted discovery of plasmid-dependent phages reveals expanded diversity of alphatectiviruses. a** Maximum likelihood tree of all known alphatectiviruses (generated with the whole genome, 14,888 sites). Branch tips in red represent the novel phages isolated in this study. All other colors (highlighted in the enlarged section of the tree) represent all previously known representatives of this phage group. **b** Map showing the site and isolation year of phages shown in (**a**). This collection includes and vastly expands the previously known diversity, despite sampling being more geographically and temporally constrained. **c** Nucleotide diversity across our collection of alphatectivirus genomes ($n = 51$). The genome map is colored to better display the nucleotide diversity value inside the gene body. Red coloration in the gene arrow symbols indicates high nucleotide diversity and blue indicates low nucleotide diversity, values correspond to the histogram above.

proposed species, we found that all 51 phages in our collection had perfectly conserved gene synteny (Fig. S1b). Just like the six previously known alphatectiviruses, they have no accessory genome and contain homologs of all 31 predicted coding genes of the PRD1 reference genome, suggesting strong constraints on genomic expansion in this group of phages. However, the isolates in our collection contain a large number of single nucleotide polymorphisms (SNPs) distributed across the entire ~15 Kb genome (Fig. 2c), and isolates ranged from 82.5% to 99% average pairwise nucleotide identity. Certain regions of the genome are highly associated with polymorphism across our collection, such as the center and C-terminus of the DNA polymerase gene, *I* (P1). Two small genes toward the end of the phage genome, *XXXVII* (P37) and *XIX* (P19), are especially associated with nucleotide polymorphisms across our genome collection. Interestingly, *XXXVII* (also called *gp v*, P37) is the outer-membrane unit of a two-component spanin system thought to be responsible for fusion of the inner and outer membrane in the final stages of cell lysis[33].

## Newly isolated ssRNA and ssDNA phages targeting IncF and IncP plasmids

In total, we isolated eight novel ssRNA phages (Fig. 1d), seven targeting the IncF plasmid and one targeting IncP. All eight ssRNA phages were related to phages in the *Fiersviridae* family of ssRNA phages and had mostly syntenic genome architectures. Analysis of the sequence of the RNA-dependent RNA polymerase or replicase, *rep*, protein homologs of the eight phages in context of other reference phage sequences showed that three of the IncF-dependent phages belonged to the *Emesvirus* genus and were closely related to phage MS2. The other four IncF-dependent ssRNA phages were related to Qbeta, in the *Qubevirus* genus. Finally, the IncP-dependent ssRNA phage, PRRIime, was closely related to phage PRR1, suggesting it is the same species and the second isolate of the *Perrunavirus* genus. Interestingly, although PRRIime was the only IncP-dependent ssRNA phage we isolated, by amino acid identity of the replicase protein it is more closely related to the MS2 phage group, than the MS2 and Qbeta groups are to each other

(Fig. 3a). None of the new ssRNA phages in our collection exhibited <80% replicase protein amino acid identity to the closest reference isolate, and therefore do not meet proposed cutoff criteria for new ssRNA phage species[34].

Finally, we isolated five new ssDNA phages targeting the IncF plasmid. All five phages were found to be related to the filamentous phage M13, within the *Inovirus* genus (Fig. 3b). One of the novel inoviruses, FfLavender, was significantly different from others in our analysis and shared 88% identity to phage M13 at the nucleic acid level across the whole genome. In line with current taxonomic guidelines, we propose that FfLavender is first isolate of the novel species *Inovirus lavender*. In general, we observed less relative diversity in the IncF-dependent phages than the IncP-dependent phages described above.

### A novel tailed phage targeting the F plasmid is related to phages targeting an orthogonal contractile pilus

The final IncF PDP we isolated, which we named FtMidnight, was found to have a 40,995 bp dsDNA genome containing putative tail genes (Fig. 3c). This finding distinguishes FtMidnight from any known PDP, which all belong to non-tailed ssRNA or ssDNA phage groups. Transmission electron microscopy confirmed that FtMidnight is a tailed phage resembling the morphological class of flexible tailed siphoviruses (Fig. 3d). To confirm the interaction of FtMidnight with the conjugation machinery of the F plasmid specifically, we sampled phage resistant micro-colonies from within FtMidnight plaques to obtain two

resistant mutants that still encoded the plasmid-borne antibiotic resistance marker. The FtMidnight-resistant mutants were collaterally resistant to F-plasmid dependent phages MS2 and Qbeta (Fig. S2c), and sequencing revealed the two mutants had independent SNPs causing a frameshift and a premature truncation of conjugation proteins TraA and TraF, respectively (Fig. S2b). Both these proteins are essential components of the conjugative pilus, suggesting that ablation of the conjugative pilus renders cells resistant to FtMidnight, and that the phage interacts directly with the pilus.

As there are no phage tail proteins known to interact with plasmid-encoded conjugation machinery, identification of the FtMidnight receptor-binding protein might have long-term applications in the development of alternative antimicrobial therapies[35]. To this end, we used structure-guided homology search to infer the probable structure and function of the FtMidnight tail proteins. Based on similarity to the distantly related marine roseophage vB_DshS-R4C[36], we identified a cluster of five proteins that are predicted to compose the distal, receptor-interacting, end of the FtMidnight tail, gp18–22 (Fig. 3c and Fig. S3a). By searching for homologs of these genes, we detected a number of siphoviral phage genomes that possess related distal-tail regions to FtMidnight (Fig. S3b). Intriguingly, many of these FtMidnight-related phages, which were isolated on hosts including *Pseudomonas*, *Xylella,* and *Stenotrophomonas*, have been documented to use the type 4 pilus (also called T4P or type IV pili) as their receptors (Fig. S3b, Supplementary Dataset 2), an orthogonal contractile pilus

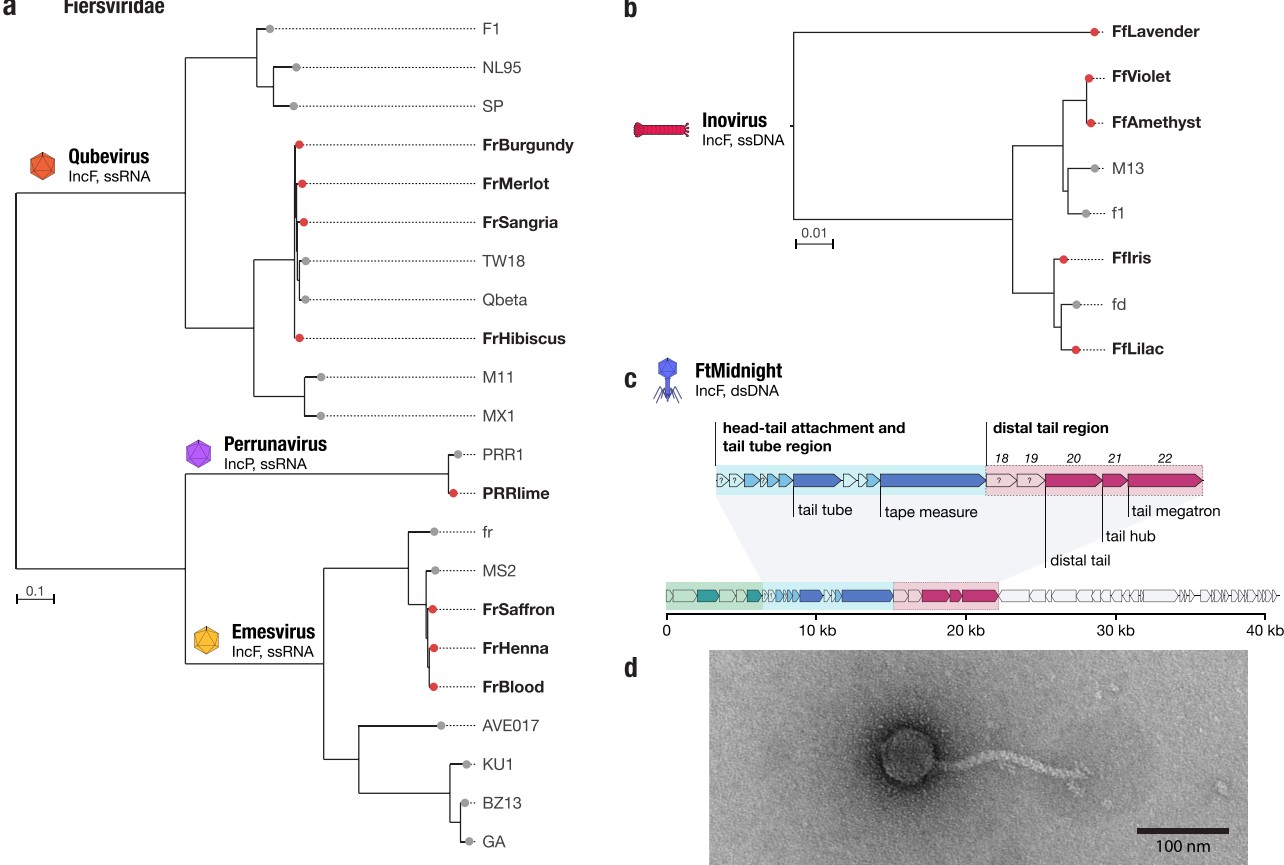

**Fig. 3 | Phage DisCo uncovers new diversity even in the best-characterized (IncF) plasmid-dependent phage system. a, b** Phylogenetic trees showing newly isolated (red branch tips) and known (gray branch tips) F dependent phages from the *Fiersviridae* family (ssRNA) and *Inovirus* (ssDNA) genera. *Fiersviridae* tree is based on the RNA-dependent RNA polymerase (635 amino acid sites), *Inovirus* tree is based on whole genome alignment (6425 nucleotide sites). **c** Genome map of the novel IncF plasmid-dependent phage, FtMidnight, highlighting structural genes.

Genes highlighted in green are predicted to be involved in head formation, the head-tail attachment and tail tube region is highlighted in blue, and distal-tail region in pink. **d** Transmission electron micrograph of FtMidnight, confirming it has siphovirus morphology (long non-contractile tail). Imaging was repeated twice independently with seven recorded observations. The image shown is representative of all observations.

that is thought to be unrelated to the conjugal pilus (a type 4 secretion system)[37]. Phylogenetic analysis of the five putative distal-tail proteins of FtMidnight implicate gp18 as most likely to be involved in receptor recognition, as it is much more divergent from homologs in T4P-associated phages than the other four proteins, and we speculate it is specifically adapted to facilitate adsorption to the F plasmid pilus (Fig. S3c).

Remarkably, both the ssRNA *Fiersviridae* and the ssDNA filamentous *Inoviridae* families include phages that use either the conjugative pilus or chromosomally encoded T4P as receptors[38,39]. Therefore, the FtMidnight-like group of phages is the third example of a phage group that can adapt to use either contractile pilus structure in short evolutionary timescales. This suggests that from a phage entry perspective, there is a high degree of functional overlap between the conjugative pilus and the T4P, which we hypothesize to be their biophysical, contractile nature.

### Plasmid-dependent tectiviruses show substantial phenotypic differences despite perfectly syntenic genomes with no accessory genes

PDP host range is dependent on plasmid host range, and therefore phages dependent on broad-host range plasmids are required to replicate in diverse host cells. Indeed, tectiviruses (alphatectiviruses) dependent on the broad-host range IncP plasmid exhibit a remarkably wide host range[40], surpassing the host breadth of any other described group of phages. This ability comes in stark contrast with their small genome size, perfectly conserved gene synteny, and lack of accessory genome. While the broad-host-range phenotype of PRD1-like phages has been long appreciated[41], the six previously isolated PRD1-like phages have been assumed to be mostly phenotypically homogeneous, perhaps due to the high level of conservation between their genomes. However, subtle differences in efficiency of plating (EoP) of some alphatectiviruses on different host strains were previously reported[42]. To explore the extent to which this constrained genomic diversity permits phenotypic variation in our larger collection of PDPs, we constructed a set of 13 hosts representing diverse Gammaproteobacteria, carrying the IncP conjugative plasmid pKJK5 (indicated by [P]). We initially observed that PDPs exhibited substantial differences in plating efficiency across hosts (Fig. 4a). For example, while PRD1 is able to plaque efficiently in all but one of the hosts, PRDcerulean can only efficiently form plaques on *Pseudomonas* hosts, representing a decrease in plaquing efficiency of at least four orders of magnitude in most other hosts. In contrast, PRDchartreuse and PRDjuniper decrease their plaquing efficiency by a similar magnitude in *P. putida*[P] when compared against *P. fluorescens*[P]. Notably, these isolates share >95% nucleotide identity to PRD1 and have no variation in gene content (Fig. S1).

We quantified host preference differences of all 51 phages on all 13 bacterial species using a high throughput liquid growth assay[43]. For each phage-host pair we calculated a liquid assay score, which represents the bacterial growth inhibition incurred by a fixed phage concentration, normalized as a percentage relative to the bacterial growth in a phage-free control (Fig. 4b). We found that, consistent with earlier plaque assays (Fig. 4a), the growth inhibition phenotype was highly variable across phage isolates (Fig. 4c). We identified more examples of phages such as PRDmint and PRDcanary that displayed a host-specialist behavior, akin to that of PRDcerulean, while others, like PRDobsidian and PRDamber appeared to robustly inhibit the growth of a wide range of hosts (host generalism). Surprisingly, when looking at the data broadly, we found that neither the phage nor the host phylogenetic relationships were strong predictors of host preference. To rule out that these host-preference differences are caused by sequence-specific anti-phage systems, we characterized the CRISPR-Cas and restriction modification (RM) systems encoded in the host genomes. We found that only four of the hosts encoded a complete

Cas operon, and that none of the spacers in the CRISPR arrays matched any of the phages in our collection (Fig. S4, Supplementary Dataset 3). We also found that all but two of the bacterial hosts harbor at least one RM system. Although interactions between plasmid-dependent tectiviruses and RM systems might play a role in host range, it cannot fully explain the differences we observe. Instead, we speculate that these host preference patterns reflect adaptation of the plasmid-dependent tectiviruses to the physiologies of different host cells. The composition of natural polymicrobial communities containing IncP plasmids likely require PDPs to rapidly adapt to infect particular assortments of taxonomically distant hosts.

### Holin protein variation contributes to host range differences in plasmid-dependent tectiviruses

To explore the genetic basis of the host-range preferences, we focused on PRDcerulean, which was the only tectivirus isolated in our narrow-host IncP plasmid screen. PRDcerulean displays the most restricted host range of our collection, and only replicates efficiently on *P. putida*[P]. On *S. enterica*[P] (and most IncP plasmid-containing hosts in our screen) PRDcerulean does not make plaques even at high titers (Fig. 4a). We reasoned that differences in host range between these phages are unlikely to be due to adsorption failure, as the receptor is encoded by an identical conjugative plasmid present in each host strain, and if there were pilus elaboration problems in any of the hosts, this should affect all phages in our collection rather than some phages individually. In line with this assumption, there was no difference in adsorption efficiency between PRD1 and PRDcerulean on *S. enterica*[P] (Fig. S5a), indicating that the replication defect in *S. enterica*[P] is receptor independent, and probably occurs once the phage chromosome has reached the interior of the cell. To understand where the replication defect occurs, we conducted high multiplicity of infection (MOI) experiments of PRDcerulean on *S. enterica*[P] in order to isolate a spontaneous escape mutant that could form plaques. We identified a single rare mutant of PRDcerulean, cer1, which was able to make plaques on *S. enterica*[P]. Sequencing revealed cer1 contained a number of mutations relative to wildtype PRDcerulean, notably in the holin gene, which encodes the protein (P35) responsible for triggering the destruction of the bacterial cell wall during cell lysis[44]. To confirm this observation, we generated a chimeric phage, cer6, by recombining the P35–P36 region of PRDcerulean with the respective sequence from PRD1. Strikingly, exchange of these two proteins restored the plating efficiency of PRDcerulean on *S. enterica*[P], although we note that the cer6 recombinant phage formed smaller plaques than PRD1 (Fig. 5a).

Analysis of the holin protein indicated that it is predicted to form three transmembrane domains (TMDs), with a short N-terminal periplasmic segment, and a longer disordered C-terminal region that extends into the cytoplasm (Fig. 5b), a topology characteristic of class I holins. The holin protein of PRDcerulean has a distinct five amino acid motif at the C-terminal end, shared only with one other phage in our collection, PRDfuschia (Fig. S5d). Despite this similarity at the C-terminus, the TMD1 and TMD2 of PRDcerulean differ significantly from those of PRDfuschia, which is able to replicate efficiently on *S. enterica*[P]. As the TMDs have been shown to be especially important for holin function[45], we hypothesized that variations in these regions could be associated with PRDcerulean's reduced host range, and we attempted to individually replace each of the regions corresponding to the TMDs of the PRDcerulean holin with the respective sequences of PRDfuschia, by recombination. Recombination with TMD1 did not yield any chimeric phages that could plaque on *S. enterica*[P], but recombinants of TMD2 yielded phages that plaqued on *S. enterica*[P] almost as efficiently as cer6 (Fig. 5a and Fig. S5b). Additionally, to recapitulate the original variant we had seen in the spontaneous mutant, we replaced the TMD3 from PRDcerulean with that of cer1. Sequencing of the resulting recombinant phages revealed that rather than recombining the entire region corresponding to the TMDs,

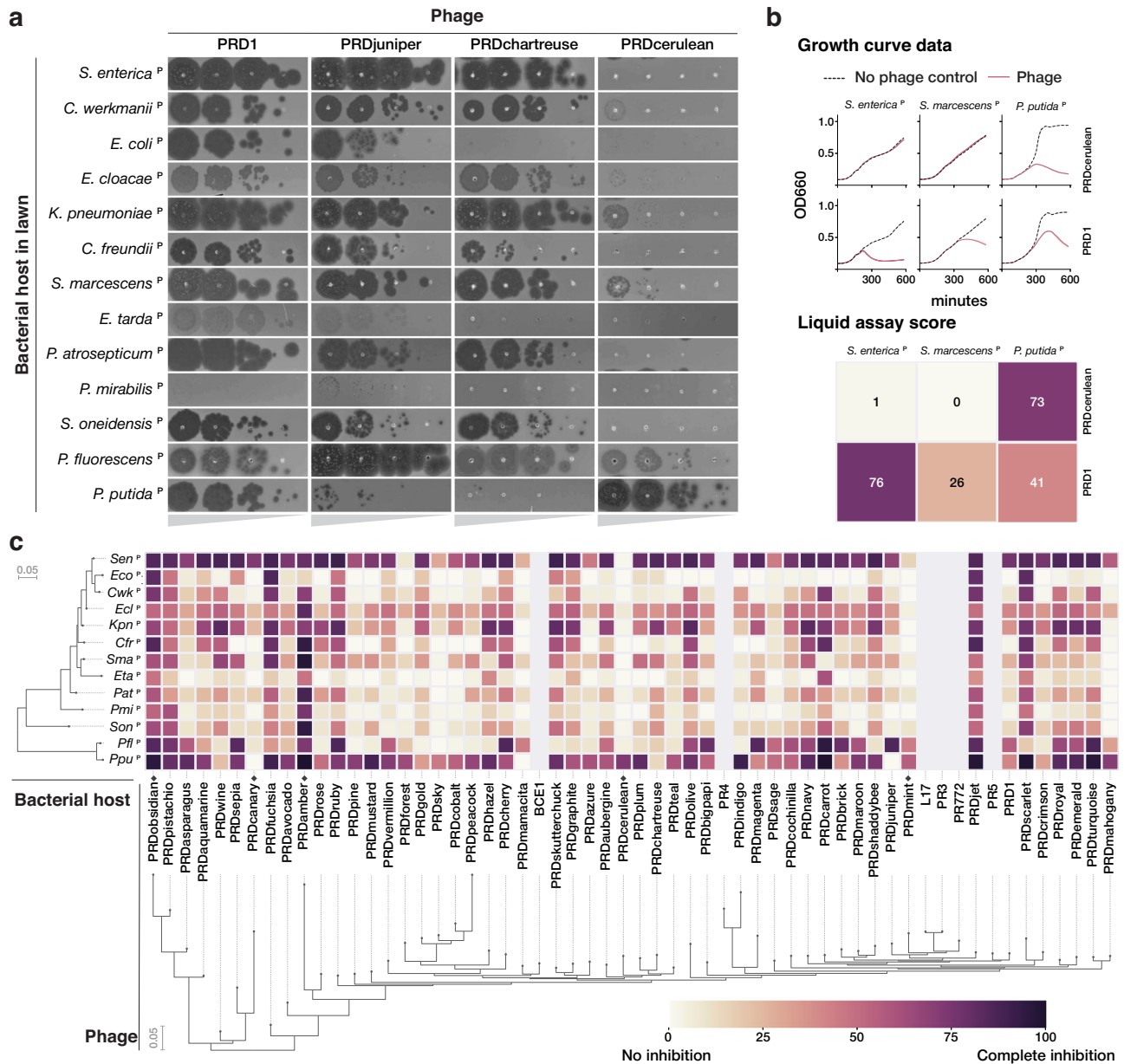

**Fig. 4 | Plasmid-dependent tectiviruses have profoundly different host range preferences. a** Plaque assays of tenfold dilutions of four plasmid-dependent tectiviruses on diverse Gammaproteobacterial hosts containing the IncP conjugative plasmid pKJK5 (indicated by ᴾ). The four phages have large differences in plaquing efficiency on different host bacteria, despite being closely related by whole genome phylogeny (Fig. 2a). **b** Top shows examples of growth curve data for phages PRD1 and PRDcerulean on three host bacteria containing the pKJK5 plasmid. Data are presented as mean values ± SEM shown as a shaded area behind the curves (error is too small to be visible). Bottom shows the same data, represented as liquid assay score. **c** High throughput estimation of host range preferences for all the novel plasmid-dependent tectiviruses in our dataset by liquid growth curve analysis. Maximum likelihood trees at the left and bottom indicate the inferred phylogenetic relationships between phages (by whole genome phylogeny) and host bacteria (by 16S phylogeny). Grayed out rows are displayed for the six published alphatectiviruses that we were unable to collect host preference data for. Black diamonds on the base of the heatmap highlight phages with host range preferences that are referenced in the text.

the phages had recombined specific SNPs from within the donor TMD sequences, allowing us to pinpoint individual variants in the PRDcerulean holin that expand host range to *S. enterica*ᴾ. Cer9 had two amino acid changes within TMD2 and cer10 had a single amino acid change inside TMD3. Furthermore, the cer10 recombinant, which was associated with poor EoP and plaque size on *S. enterica*ᴾ, proved to be unstable in culture, and larger plaques spontaneously appeared after one round of replication. Re-sequencing of the larger plaque mutant, cer11, revealed it had acquired a frameshift mutation in the C-terminal end of the protein, which reverts the C-terminal motif close to the longer motif found in most variants in our collection, e.g.,

PRDaquamarine (Fig. S5e). This indicates that the mutation in the C-terminal cytoplasmic end of the protein might have an epistatic interaction with the TMD mutations.

Mapping of these mutations onto the holin membrane topology prediction showed that they are clearly located within membrane-embedded portions of the holin protein, and further AlphaFold modeling of the holin secondary structure indicated that the TMD2&3 mutations may be spatially proximal in the native protein structure (Fig. S5c). Overall, these results indicate that one of the largest hurdles for plasmid-dependent tectiviruses to achieve infection of diverse bacterial hosts may be adapting the phage lysis components to various

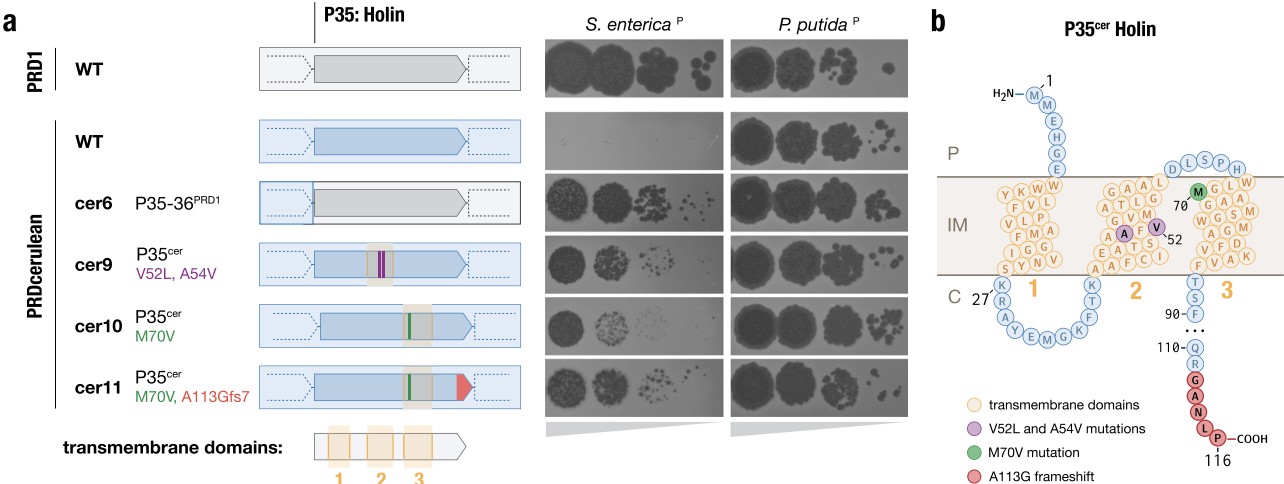

**Fig. 5 | Differences in the holin protein explain reduced host range of PRDcerulean. a** Schematic representation of holin (P35) mutants, and their effect on plaque formation. Gray arrows represent the WT PRD1 holin gene, blue arrows represent the WT PRDcerulean holin gene. Amino acid changes are represented by purple, green, and red marks on the holin gene body. To the right, the corresponding plaque assays of tenfold dilutions on *S. enterica*[P] and *P. putida*[P] are shown. Complete replacement of the holin gene, as well as specific mutations in the TMD2&3 restore the plating efficiency of PRDcerulean on *S. enterica*[P]. **b** Diagram of the predicted membrane topology of the holin protein (P35[cer]). It is predicted to have a short N-terminal periplasmic segment, three transmembrane domains (TMDs), and a longer disordered C-terminal region that extends into the cytoplasm. P = periplasm, IM = inner membrane, and C = cytoplasm. Relevant mutations are colored as in (**a**).

host cell physiology, e.g., inner membrane composition. We speculate that the PRDcerulean lysis proteins may be specifically adapted to work in Pseudomonad hosts, and note that though the phage appears to have a narrow-host range in our limited screen, we may simply not be testing the natural hosts of PRDcerulean. Notably, none of the holin mutations affected replication on *P. putida*[P] (Fig. 5a and Fig. S5b). The finding that altered host range is accessible within a small number of point mutations suggests there is immense functional flexibility encoded within the proteins of plasmid-dependent tectiviruses, and while there appears to be a strict constraint on genome size, these viruses may acquire accessory function through protein variation rather than gene gain, in contrast to canonical tailed phages which are associated with extensive mosaicism.

## Metagenomic approaches fail to recover plasmid-dependent tectiviruses

Given the small number of plasmid-dependent tectiviruses known prior to this study (6, excluding BCE1) we were surprised by how readily discoverable these phages were in our samples (though we note that low numbers of characterized representative phages does not necessarily reflect low environmental abundance). To quantify their absolute abundance, we used Phage DisCo to estimate the concentration of IncP PDPs in fresh influent from two wastewater sites in Massachusetts, USA, relative to species-specific phages of *E. coli, S. enterica*, and *P. putida* (Fig. 6a). Phages dependent on the IncP plasmid RP4 were present in wastewater at ~1000 phages/mL, the same order of magnitude as species-specific phages of *E. coli* at ~4000 phages/mL. Species-specific phages of *S. enterica* and *P. putida* were less abundant than IncP-PDPs, present at ~100 phages/mL and ~5 phages/mL, respectively. While this absolute quantification is limited by the use of a single strain to capture all species-specific phages, PDP quantification may be similarly limited by use of a single plasmid and two host species to capture all IncP PDPs. Nevertheless, wastewater is considered one of the best samples in which to find *E. coli* and *S. enterica* phages, and therefore despite the limitations of this relative abundance metric, the data show that these phages are common, at least in built environments (human-made environments). The extent to which this abundance is a characteristic of phages dependent on IncP-type plasmids as opposed to PDPs in general remains to be seen, although these estimates are in line with reports of

phage abundance for multiple different plasmid types in wastewater from Denmark and Sweden[26].

Metagenomic-based viral discovery techniques have been extremely successful in expanding known viral diversity[46–48]. Although some studies have identified tectiviruses in metagenomic datasets[49] and metagenomic-assembled genomes[50], alphatectiviruses have yet to be found in metagenomic analyses, at odds with the relatively high abundance of the plasmid-dependent alphatectiviruses in wastewater (Fig. 6a). With the increasing availability of metagenomic datasets, we decided to reexamine the presence of this group of phages in assembled metagenome collections. We queried the JGI IMG/VR database of uncultivated viral genomes (UViGs) and retrieved genomes with a match to the Pfam model PF09018, which corresponds to the PRD1 coat protein, which is conserved across all known tectiviruses. This search retrieved a set of diverse genomes in which, using refined models built from our alphatectivirus collection, we identified homology to diagnostic tectivirus proteins[14], such as DNA polymerase (P1), DNA packaging ATPase (P9), and tail tube DNA translocation proteins (P18, P32) in addition to the coat protein (P3) used for the retrieval of these sequences (Fig. S6b). However, none of the UViGs appear to belong to any of the pre-existing groups of isolated tectiviruses (Fig. S6c) suggesting there is large unexplored diversity in the *Tectiviridae* family.

We tested if we could recover alphatectivirus sequences through metagenomic sequencing of our own wastewater samples, where we knew these phages were present at high abundance (around 1000 PFU/mL) (Fig. 6a). We processed our samples by filtration, and further concentrated the viral fraction by 100-fold, before performing DNA extraction and bulk sequencing (average of 2M reads per sample). We classified our metagenomics dataset with Kraken2 and found that a very small proportion of the reads (<0.001%) could be assigned to the *Alphatectivirus* taxonomic group, which would not be sufficient for assembly (Fig. 6b). This implied that, despite there being no assembled alphatectiviruses in public databases, they may still be identifiable in raw reads.

We then looked at additional published wastewater metagenomic sequencing datasets, and processed samples from diverse projects, representing different sequencing depths, locations, and sample processing methods, comprising a total of 290 samples and more than

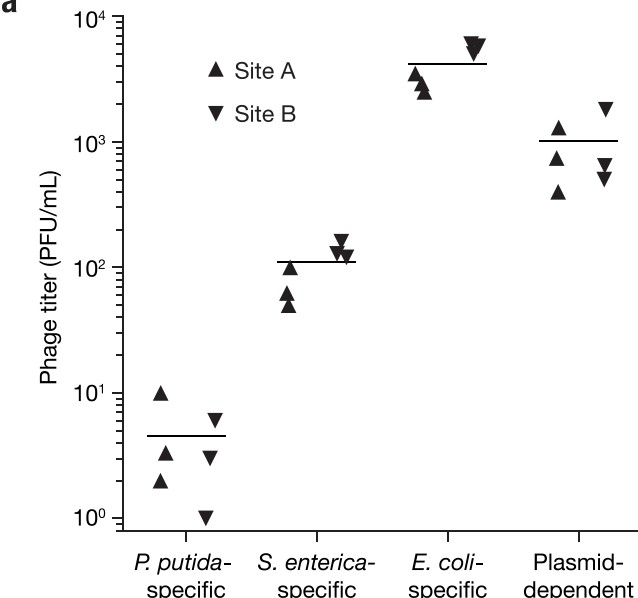

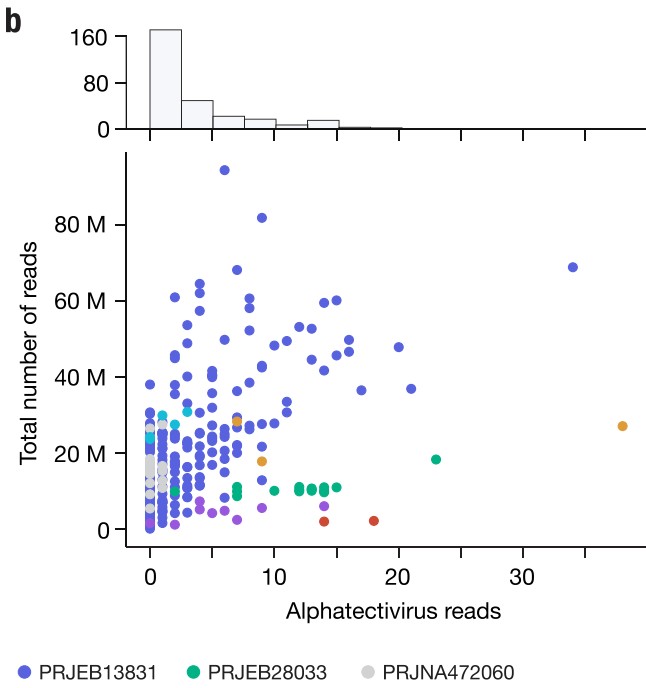

**Fig. 6 | Alphatectiviruses are underrepresented in metagenomic assembled viromes. a** Abundance of plasmid-dependent phages in wastewater influent. Plasmid-dependent phages targeting the IncP plasmid RP4 are orders of magnitude more abundant than *P. putida*- and *S. enterica*-specific phages, in two independent wastewater influent samples. *n* = 2 biologically independent samples, measured in three technical replicates. **b** Histogram and scatter plot of reads classified as being of alphatectiviral origin, against total number of reads in each metagenomic sample analyzed. Colors indicate different BioProjects from the SRA, full metadata can be found in Supplementary Dataset 2.

5 billion reads total (Supplementary Dataset 2). Over 75% of the samples contained five or fewer reads assigned to alphatectiviruses (Fig. 6b). However, we found some alphatectivirus reads, primarily from the larger datasets, which directly mapped to the PRD1 reference genome (Fig. S6a). The recovered reads appeared to be bona fide alphatectivirus sequences, as shown by the high mapping quality to

the reference, a conservative approach that would fail to identify isolates with higher variation. Taken together, no single dataset we analyzed contains enough reads to assemble a complete alphatectivirus genome. We hypothesize that a combination of a low relative abundance, small genome size, and highly polymorphic population might be responsible for the absence of alphatectiviruses in metagenomic assembled genome collections. Overall, this finding points to a discordance between culture-based and metagenomic-based virus surveillance.

## Discussion

Our finding that phages exploiting conjugative plasmid-encoded receptors are common and abundant in the urban environment suggests that PDPs act as an important and underappreciated constraint on the spread of conjugative plasmids. Though studies have shown that conjugative plasmids can rapidly evolve resistance to PDPs[17,51], these studies also suggest that resistance comes with a tradeoff in conjugation efficiency, such that phage-resistant plasmids cannot easily spread to new hosts. This suggests that with further study and discovery, PDPs could be exploited to manipulate the dynamics of conjugative plasmid mobility, and thus the spread of antibiotic resistance genes in high-risk environments. PDPs may be particularly applicable to controlling epidemics of plasmid acquired resistance, for example the current epidemic of carbapenem-resistant Enterobacteriaceae mobilized by IncX3 conjugative plasmids[52–54].

A challenge in the potential translational application of PDPs may be ensuring that phage host range is sufficiently broad to avoid the formation of plasmid reservoirs in bacterial hosts that cannot be infected by phages. Our finding that plasmid-dependent tectiviruses have highly variable host range preferences reinforces the significance of this hurdle. However, our investigation into the host range of these phages showed that this phenotype is, to some extent, genetically encoded in the phage lysis machinery. Further study is necessary to better understand the genetic basis of host range in plasmid-dependent tectiviruses and PDPs more broadly, but the expansion of host range of PRDcerulean via gene exchange may be an exciting step toward predicting and engineering the host range of these phages. From a virus evolution perspective, this finding illustrates the great functional flexibility contained within PDP lysis proteins, which we speculate may be necessary for rapid adaptation to new host cells as their associated conjugative plasmids transmit across communities of diverse bacteria.

Our discovery of FtMidnight, along with the significant expansion of other known conjugative PDP families, highlights the power of Phage DisCo to uncover new phage diversity. A more comprehensive understanding of the diversity of PDPs may shed light on outstanding questions as to the evolution of plasmid-dependency in phages. Indeed, our discovery that the F pilus-dependent phage FtMidnight is related to type 4 pilus targeting phages suggests that there may be some functional, if not evolutionary, relationship between these purportedly unrelated structures. It remains to be seen how this finding translates to other groups of PDPs. For example, only the *Alphatectivirus* genus within the broader *Tectiviridae* family are known to depend on plasmid-encoded receptors, and the receptors of other genera, such as the betatectiviruses that infect *Bacillus* species, are thought to be components of the cell wall[55], although we note there is very little similarity between the spike proteins of PRD1 and the most characterized *Betatectivirus* Bam35[56]. Likewise, a recent study identified a new pair of short tailed phages that are dependent on conjugative plasmids belonging to the IncN group[57] but the receptor dependency of phages with related tail proteins is unknown. Further study of such viruses that are evolutionarily adjacent to plasmid-dependent groups may reveal parallel evolutionary routes to plasmid-dependency. Additionally, further characterization of the diversity of phage receptor-binding proteins that interact with plasmid-encoded pili

 9

could eventually facilitate the engineering of plasmid-targeting phenotypes into genetically engineered phages or phage-derived particles, which may offer long-term promise as alternative antimicrobial therapies[35].

The relatively high abundance of IncP PDPs in wastewater as measured by culture-based methods contrasts with their absence from metagenomic datasets, indicating a blind spot in bulk sequencing based approaches to detect certain groups of viruses. The biochemical properties of some viruses have been suggested to play a role in their depletion from metagenomic datasets, such as DNA genomes with covalently bound terminal proteins[58]. Though we cannot rule out a similar phenomenon is responsible for the lack of plasmid-dependent tectiviruses in some metagenomic samples, our metagenomic extractions were protease treated yet had comparable abundance of plasmid-dependent tectiviruses relative to public datasets. We speculate that other factors might play a role, including the small genome size of PDPs relative to other viruses, low relative abundance compared to other viruses, and high within sample sequence diversity interfering with consensus-assembly based methods. Consistent with our observations, high strain heterogeneity has previously been shown to hinder metagenomic assembly of abundant marine viruses[59], and benchmarking studies with simulated metagenomic data has found this to be an intrinsic limitation of both viromic and metagenomic sequencing studies[60,61]. These discrepancies point to the continued need for systematic culture-based viral discovery and method innovation.

Though we chose to focus this initial study on conjugative plasmids that are already known to be targeted by PDPs, we anticipate that the Phage DisCo method will be generally applicable to identifying phages dependent on other conjugative plasmid systems, as well as translatable to further specialized phage discovery screens. The diversity and abundance of the PDPs we detected in the urban environment leads us to hypothesize that the interplay between phages and conjugative plasmids, both selfish genetic elements, may be driving the diversification of the conjugation systems mediating horizontal gene transfer in bacteria. This work represents a major first step in the large-scale exploration of this functional group of phages, and much remains to be discovered about their ecology and biology, including how they interact with the plethora of defense systems present in bacteria[62].

## Methods

### Strains and growth conditions
Details of all bacterial strains, plasmids, phages, and primers used and constructed in this study are available in Supplementary Dataset 1a–d. Unless stated otherwise, bacteria were grown at 37 or 30 °C in autoclaved LB$^{Lennox}$ broth (LB: 10 g/L Bacto Tryptone, 5 g/L Bacto Yeast Extract, 5 g/L NaCl) with aeration (shaking 200 rpm) or on LB agar plates, solidified with 2% Bacto Agar at 37 or 30 °C. Salt-free LBO media contained 10 g/L Bacto Tryptone, 5 g/L Bacto Yeast Extract. When required antibiotics were added at the following concentrations: 50 µg/mL kanamycin monosulfate (Km), 100 µg/mL ampicillin sodium (Ap), 20 µg/mL tetracycline hydrochloride (Tc), 30 µg/mL trimethoprim (Tm), 20 µg/mL chloramphenicol (Cm), and 20 µg/mL gentamicin sulfate (Gm).

### Phage replication
Replication host strains for all phages used in this study are detailed in Supplementary Dataset 1c. High titer phage stocks were produced by adding ~10$^5$ Plaque Forming Units (PFU) to exponential phase cultures at ~OD$_{600}$ 0.1, and infected cultures were incubated for at least 3 h at 37 °C (with aeration). Phage lysates were centrifuged (10,000 × $g$, 1 min) and supernatants were sterilized with 0.22 µm filters. Phage lysates were serial-diluted (decimal dilutions) with SM buffer and PFU enumeration was performed by double-layer overlay plaque assay[63], as follows. Bacterial lawns were prepared with stationary phase cultures

of the host strains, diluted 40 times with warm top agar (0.5 % agar in LB, 55 °C). The seeded top agar was poured on LB 2% agar bottom layer: 3 mL for 8.6 cm diameter petri dishes or 5 mL for 8.6 × 12 cm rectangular petri dishes. When required, antibiotics were added to the top agar at concentrations specified above.

### Plasmid construction
The F plasmid from strain SVO150 was modified via recombineering to encode a *gfp* locus and kanamycin resistance locus (*aph*) for selection (FΔ*finO*::*aph*-P*lac*-*gfp*) to aid in conjugation and rapid identification of plasmid⁺ colonies. Briefly, SVO150 was electroporated with the pSIM5tet recombineering plasmid (Supplementary Dataset 1b), and the native IS3-interrupted *finO* locus was replaced with the *aph*-P*lac*-*gfp* cassette from pKJK5 using primers NQO2_9 and NQO2_12. The replaced region was amplified with primers NQO2_5 and NQO2_6 and sent for Sanger sequencing to confirm the correct replacement.

### Strain construction
For differential identification of plaques in coculture and transconjugant selection, constitutive *sgfp2\** or *mScarlet-I* loci along with a chloramphenicol resistance locus were added to *E. coli*, *S. enterica,* and *P. putida* strains (Supplementary Dataset 1a). Tn7 transposons from pMRE-Tn7-145 and pMRE-Tn7-152 were introduced into the *att$^{tn7}$* site via conjugation from an auxotrophic *E. coli* donor strain as previously described[64].

The RP4 plasmid was introduced into chromosomally tagged *S. enterica* and *P. putida* via conjugation using the BL103 donor strain. Overnight liquid cultures of donor and recipient strains were mixed at a 1:10 (donor:recipient) ratio and concentrated into a volume of 20 µl by centrifugation. The cell slurry was transferred to the top of a 12 mm, 0.45 µm nitrocellulose membrane on the surface of an LB agar plate for 4 h at temperature optimal for the recipient strain (see Supplementary Dataset 1a) to permit conjugation. Transconjugants were selected by plating on LB supplemented with chloramphenicol and kanamycin. For FΔ*finO*::*aph*-P*lac*-*gfp*, a plasmid and prophage-cured *S. enterica* strain (SNW555, D23580 ΔΦ ΔpSLT-BT ΔpBT1 ΔpBT2 ΔpBT3[65]) was used to mitigate any interference from the IncF *Salmonella* virulence plasmid (pSLT) and native prophages. The FΔ*finO*::*aph*-P*lac*-*gfp* plasmid was introduced into SNW555 and NQO62 via conjugation, exactly as described above.

For IncP-PDP host range experiments, the pKJK5 plasmid was transconjugated into *P. putida* KT2440, *Pectobacterium atrosepticum* SCRI1043, *Shewanella oneidensis* MR1, *Serratia marcescens* ATCC 1388, *Enterobacter cloacae* ATCC 13047, *Pseudomonas fluorescens* Pf0-1, *Klebsiella pneumoniae* PCI 602, *Citrobacter werkmanii* IC19Y, *Citrobacter freundii* ATCC 8090, *Edwardsiella tarda* ATCC 15947, *Proteus mirabilis* BB2000 Δ*ugd*(immobile mutant), and *S. enterica* serovar Typhimurium LT2 via the cross streak method. The pKJK5 plasmid contains *gfp* under the control of the P*lac* promoter, which results in derepressed fluorescence in non-*E. coli* (*lac* negative) hosts[66]. Additionally, the pKJK5 donor strain, NQO38, constitutively expresses *mCherry*, permitting easy identification of transconjugants without need for dual selection. Briefly, an overnight liquid culture of the donor strain NQO38 was applied vertically in a single streak down the center of an LB agar plate. Subsequently, an overnight liquid culture of a recipient strain was streaked horizontally across the plate, crossing over the donor streak. After incubation at the recipient optimal temperature, transconjugant colonies were purified on the basis of green fluorescence signal.

### Optimization of PDP detection by fluorescence-enabled coculture
To validate the use of fluorescence-enabled coculture to detect PDPs, a *S. enterica*-specific phage (9NA), a *P. putida*-specific phage (SVOΦ44), and an IncP PDP (PRD1) were mixed at equal concentration

(-10³ PFU/mL). In total, 100 μL each of overnight liquid cultures of *S. enterica* LT2 *att*$^{Tn7}$*::Tn7-mScarlet-I* + RP4 (NQO89) and *P. putida* *att*$^{Tn7}$*::Tn7-SGFP2* * + RP4 (NQO80) was added to 3 mL molten LB top agar, along with 10 μL of the phage mixture, and poured onto an LB agar plate. Plates were incubated overnight at 30 °C and then imaged in brightfield, red fluorescence channel, and green fluorescence channel using a custom imaging platform.

The custom imaging setup has a Canon EOS R camera with a Canon 100 mm lens with LEDs paired with excitation and emission filters (Green: 490–515 nm LED with 494 nm EX and 540/50 nm EM filters; Red: 567 nm LED with 562 nm EX and 641/75 nm EM filters). Excitation filters are held in a Starlight Xpress emission filter wheel. The camera, LEDs, and filter wheel are all controlled with custom software. Exposure times were 0.25 [green] and 0.5 s [red], with camera set to ISO-200 and f/3.5 as experimentally determined to maximize dynamic range. Imagining parameters were selected such that when green and red fluorescence channel images were merged, all three phages could be easily identified by fluorescent plaque phenotype: 9NA phages were visible as green plaques (only *P. putida att*$^{Tn7}$*::Tn7-SGFP2* * + RP4 grows in these areas), SVOΦ44 plaques were visible as red plaques (only *S. enterica* LT2 *att*$^{Tn7}$*::Tn7-mScarlet-I* + RP4 grows in these areas) and PRD1 plaques had no fluorescent signal (neither species grew in these areas). The red and green channels were separated from their raw images, their exposure linearly rescaled, and remapped to the red and blue channels respectively (to enhance visual color contrast). All image manipulations were done with scikit-image v0.17.2[67].

## Collection and processing of environmental samples
For phage isolation, wastewater primary influent from a total of four sites in Massachusetts were collected, along with soil, animal waste, and compost from farms, community gardens and parks close to Boston, USA. Sample collection details can be found in Supplementary Dataset 1e (Environmental Samples). All samples were resuspended (if predominantly solid matter) in up to 25 mL of sterile water and incubated at 4 °C for 12 h with frequent vortexing to encourage suspension and homogenization of viral particles. The resuspended samples were centrifuged at 4000 × *g* for 30 min to pellet large biomass, and the clarified supernatant was filter sterilized using a 0.22 μm vacuum driven filtration unit to remove bacteria. Filtered samples were stored at 4 °C. For metaviromic sequencing and phage enumeration in wastewater influent, two 100 mL samples were collected in September 2022 from two separate intake sources of wastewater at a treatment plant in Boston, MA. Samples were processed by filtration as described above, except that processing was initiated immediately upon sample collection to avoid any sample degradation.

## Isolation of novel environmental PDPs by fluorescence-enabled coculture
For high throughput discovery of PDPs targeting the IncP plasmid pilus, co-culture lawns of *S. enterica* LT2 *att*$^{Tn7}$*::Tn7-mScarlet-I* + RP4 (NQO 89) and *P. putida attTn7::Tn7-SGFP2* * + RP4 (NQO80) were prepared as described earlier, except that 100 μl of filtered environmental samples containing putative novel phages were added instead of the reference phages. In cases where phage load in samples was too high, and subsequent lawn did not grow uniformly due to widespread lysis, the amount of filtered sample added to the lawns was diluted 10-fold until single plaques were obtained. Putative PDP plaques (exhibiting no fluorescence) were sampled using sterile filter tips, diluted and re-plated for single plaques at least twice to ensure purity. For the IncF plasmid-targeting phages, the procedure was the same, except that strains SVO348 (*E. coli* MG1655 *att*$^{Tn7}$*::mScatlet-I-gmR* + F*ΔfinO::aph-gfp*) and NQO87 (*S. enterica* D23580 ΔΦ ΔpSLT-BT ΔpBT1 ΔpBT2 ΔpBT3 + F*ΔfinO::aph-gfp*) were used in the lawns. The plasmid and prophage-cured strain of *S. enterica* was used for the IncF-dependent phage screen to mitigate interference from the native *Salmonella* virulence plasmid (which belongs to incompatibility group F[68]) and prophages.

Once putative novel PDPs had been purified from environmental samples, 5 μl drops of tenfold dilutions were plated on lawns of isogenic plasmid-free host strains (BL131, SVO126, SVO50, or SNW555) to confirm plasmid-dependency. We note that false positives (i.e plasmid independent phages that infected both species in the coculture) were occasionally obtained during the IncF PDP isolation, due to the phylogenetic proximity between *E. coli* and *S. enterica*, suggesting that use of more distinct host strains (if possible for the plasmid of interest) maximizes assay efficiency.

## Phage DNA and RNA extraction and sequencing
Pure phage stocks that had undergone at least two rounds of purification from single plaques and had titers of at least 10⁹ PFU/mL were used for nucleic acid extraction. The Invitrogen Purelink viral RNA/DNA mini kit was used to extract genetic material from all phages according to manufacturer instructions. High absorbance ratios (260/280) of 2.0–2.2 were considered indicative of RNA phage genomes. To remove host material contamination, putative RNA samples were incubated with DNase I (NEB) for 1 h at 37 °C and inactivated afterwards with EDTA at a final concentration of 5 mM. RNA was reverse transcribed using SuperScript™ IV VILO™ (Invitrogen™) for first strand synthesis, per the manufacturer's instructions. Second strand synthesis was performed by incubating the cDNA with DNA Ligase, DNA Polymerase I, and RNase H in NEBNext® Second Strand Synthesis Reaction Buffer (NEB) at 16 °C for 3 h. cDNA was then used in downstream library preparation. Additionally, as all known non-RNA IncF PDPs have ssDNA genomes which are incompatible with tagmentation-based library preparation, any putative DNA sample from IncF PDPs was subjected to second strand synthesis as described above. Illumina sequencing libraries of the DNA and cDNA samples were prepared as previously described[69]. Sequencing was carried out on the Illumina Novaseq or iSeq to produce 150 bp paired end reads. To improve the assembly quality of the RNA phage genomes, we conducted a second round of sequencing of the same RNA samples using the NGS provider SeqCoast Genomics. RNA samples were prepared for whole genome sequencing using an Illumina Stranded Total RNA Prep Ligation with Ribo-Zero Plus Microbiome and unique dual indexes. Sequencing was performed on the Illumina NextSeq2000 platform using a 300 cycle flow cell kit to produce 150 bp paired reads. The genetic composition (dsDNA vs ssDNA) for phage FtMidnight was inferred via fluorescence signal using the Quant-IT dsDNA kit (Invitrogen).

For metaviromic DNA extraction, 45 mL of freshly filtered influent from each of the two extraction sites was concentrated 100× into 500 μl using 100 kDa molecular weight cutoff centrifugal filter units (Amicon). Nucleic acids were extracted from 200 μl of concentrated filtrate, and sent to SeqCenter for library preparation and Illumina sequencing. Sample libraries were prepared using the Illumina DNA Prep kit and IDT 10 bp UDI indices, and sequenced on an Illumina NextSeq 2000, producing 2 × 151 bp reads.

## Phage genome assembly and annotation
Sequencing reads were adapter trimmed (NexteraPE adapters) and quality filtered with Trimmomatic v.0.39[70]. For samples with very high read depth, filtered reads were subsampled with rasusa v.0.5.0[71] to an ~200× coverage to facilitate assembly. The reads were then assembled with Unicycler v.0.4.8[72] or rnaviralSPAdes v.3.15.5[73]. The annotations from curated PRD1, MS2, Qbeta, and M13 reference genomes were transferred to the resulting assemblies with RATT v.1.0.3[74] and manually curated for completion. Phage isolates with redundant genomes were removed from the analysis and all phages included in this study represent unique isolates. Reads are deposited in the NCBI Sequence Read Archive (SRA). All accession numbers for previously published

genomes and those generated in this study are listed in Supplementary Dataset 1c.

## Nucleotide diversity

To calculate nucleotide diversity among the alphatectiviruses, all the assembled isolates were aligned to the PRD1 reference genome with minimap2 v2.24[75]. Resulting alignments were processed with bcftools v1.9[76] and samtools v1.6[77] to then calculate nucleotide diversity with vcftools v0.1.16[78] with a sliding window of size 100 bp. Results were plotted with seaborn v0.12.2[79] and matplotlib[80]. Novel species classifications for alphatectiviruses were proposed where average pairwise nucleotide diversity was less than 95%[30].

## Phage enumeration in wastewater by plaque assay

Two freshly filtered wastewater influent samples were processed as previously described (see "Collection and processing of environmental samples") and the concentration of phages in volumes of 10, 100, and 500 μm were enumerated by single-host plaque assay on strains SVO50, BL131, and SVO126 and by fluorescence-enabled co-culture plaque assay on NQO89 and NQO80. All phage enumeration was performed with three biological replicates. Titers per milliliter were calculated and plotted for both sites.

## Determination of phage host range

Host range of the IncP-PDPs was assessed by traditional EoP assay or by killing in liquid culture by $OD_{660}$ measurement, based on a previously described method[43]. All the phages were challenged against the following bacteria containing the pKJK5 plasmid: *P. putida* KT2440, *Pectobacterium atrosepticum* SCRI1043, *Shewanella oneidensis* MR1, *Serratia marcescens* ATCC 1388, *Enterobacter cloacae* ATCC 13047, *Pseudomonas fluorescens* Pf0-1, *Klebsiella pneumoniae* PCI 602, *Citrobacter werkmanii* IC19Y, *Citrobacter freundii* ATCC 8090, *Edwardsiella tarda* ATCC 15947, *Proteus mirabilis* BB2000 *Δugd*, and *S. enterica* serovar Typhimurium LT2. These hosts were chosen as they all showed some degree of susceptibility to IncP-dependent phages when transconjugated with the pKJK5 plasmid, indicating proper elaboration of the IncP pilus.

For the high throughput determination of host range, phages were normalized to a titer of $10^7$ PFU/mL as measured in strain NQO36, with the exception of PRDchartreuse, PRDcanary, PRDjuniper, and PRDmamacita, which were normalized to the same titer in NQO37, due to their inability to replicate to high titers in NQO36. Growth curve experiments were set up in 96-well plates with each well containing 180 μL of bacterial culture at $OD_{600}$ of ~0.1 and 20 μL of phage stock when appropriate, for a final concentration of $10^6$ PFU/mL. They were grown in a plate reader (Tecan Sunrise™) for 10 h with shaking, at the optimal temperature for the strain (see Supplementary Dataset 1a), measuring the optical density at 660 nm, every 5 min. Each 96-well plate had a phage-free control, cell-free control, and the strain-phage condition in triplicate. To calculate the liquid assay score of each host-phage pair we followed the method described previously[43]. Briefly, we calculate the area under the growth curve for each host-phage pair, as well as for its corresponding phage-free control grown in the same plate. The mean area under the curve value is then normalized as a percentage of the mean area under the curve in the phage-free control. Growth curves are plotted with shading representing the standard error. Liquid assays scores are plotted as a heatmap, and are vertically sorted according to the previously computed alphatectivirus tree and horizontally sorted according to a 16S tree of the bacterial hosts (see Supplementary Dataset 1a).

## Adsorption assay

In total, 50 μl of exponentially growing SVO126 and NQO37 cells at a density of ~$10^8$ CFU/ml were mixed in a 96-well plate with 50 μl of PRD1 or PRDcerulean at a density of $10^6$ PFU/ml to achieve an MOI of ~0.01.

Adsorption was done in triplicate for each strain-phage combination, and cell-free media controls were used in place of cells to quantify the maximum unadsorbed concentration of phage. After 10 min adsorption time at 37 °C, the 96-well plate containing the cell-phage mixtures was mounted on top of a sterile 96-well MultiScreenHTS GV Filter Plate with sterile 0.22 μm membrane (Millipore) and centrifuged at $4000 \times g$ to remove cells and adsorbed phages. Unadsorbed phage was quantified by serial dilution of the filtrate and plaque assay as described in Phage Replication above. Unadsorbed phage were represented as a percent of the maximum unadsorbed phage derived from cell-free media controls.

## Phage recombination

To replace the holin gene of PRDcerulean, *S. enterica*-RP4 with recombineering plasmid pSIM5tet (SVO296) was used. Briefly, bacteria were grown to exponential phase in LB at 30 °C, with selective antibiotics for both plasmids, as specified in Supplementary Dataset 1b. The culture was then infected with high titer PRDcerulean lysate to a final concentration of ~$10^7$ PFU/mL for 15 min. The culture was then induced for recombination for 15 min at 42 °C. Electro-competent cells were then prepared by cooling the cells for 10 min followed by three washes with cold sterile Mili-Q water at a 1:1 volume, and concentrated 50 times in cold sterile Mili-Q water. Competent cells were then mixed with ~100 ng of DNA in 1 mm gap cuvettes and electroporated (1.8 kV, 25 μF, 200 Ω). Electroporated bacteria were mixed with 100 μL of fresh overnight liquid culture of *S. enterica* RP4 (NQO89) and the mixture was plated in a double-layer overlay plaque assay as described above. Only successful recombinant phages formed plaques on the bacterial lawn, and those where isolated and purified for further analysis. The holin gene DNA substrates for recombination were obtained with primers NQO3_13–NQO3_20.

## Holin structure prediction

The topology of the PRDcerulean holin protein (P35cer) was predicted with the CCTOP v1.1.0 web server[81] and drawn with Protter v.1.0[82]. The structure was predicted with ColabFold v1.5.3[83], and rendered with PyMol v.2.5.6[84]. Model parameters are specified in the code repository. The holin multiple sequence alignment was generated with clustalo v1.2.4[85] and visualized with UGENE v.38.1[86].

## FtMidnight-resistant mutants

To isolate mutants of the F-plasmid that were spontaneously resistant to FtMidnight, a dilution series of phage was plated on SVO348, with kanamycin in the top agar in order to select against phage resistance via plasmid loss (the F plasmid derivate FΔ*finO::aph-Plac-gfp* contains a kanamycin resistance marker), as already described in Phage Replication above. Plates were incubated for 24 h at 37 °C, and then a further 24 h at room temperature, after which phage resistant micro-colonies were visible within the zones of phage lysis. Two independent colonies were picked and restreaked onto LB kanamycin agar. Restreaking was repeated once more to ensure purity. To understand resistance phenotype, the mutants were screened by plaque assay for susceptibility to FtMidnight, MS2, and Qbeta. To find the causative mutations, genomic DNA was extracted using the Quick-DNA Miniprep Plus Kit (Zymo) according to manufacturer instruction, and sequenced to >30× genome coverage with Nanopore technology using v14 library preparation chemistry and an R10.4.1 flow cell by the provider Plasmidsaurus' bacterial genome sequencing service.

## FtMidnight structural annotation and homology search

The genome of FtMignight was originally annotated with prokka v1.14.6[87] using the PHROGs database[88]. Annotations were manually curated to identify specific structural components by template-based homology search against the PDB_mmCIF70 database with HHpred through the MPI Bioinformatics Toolkit[89]. The structure model for

FtMidnight was based on these structural hits, which can be found in Supplementary Dataset 2. A list of phage genomes containing homologs of the gp18-gp22 proteins was collected by searching with the tblastn[90] web server against the nucleotide collection. A selection of phages with a conserved distal-tail region were visualized with clinker[91], and the receptor was annotated if found in the literature. Accessions and references can be found in Supplementary Dataset 2.

## Annotation of CRISPR-Cas and RM in bacterial genomes
CRISPR-Cas systems and spacers were annotated with CRISPRCasTyper v1.8.0[92], and RMs were annotated with DefenseFinder v1.0.9[93]. All the spacers were then searched with blastn v2.15.0[94] against the complete alphatectivirus genomes, but no hits were recovered from this search. All the Accessions to bacterial genomes can be found in Supplementary Dataset 1a, and results of this search are included in Supplementary Dataset 3.

## Search and comparison of tectiviruses in metagenomic assembled genomes
To collect metagenomic assembled genomes of tectiviruses, a search was performed in the JGI IMG/VR[95] for UViGs matching Pfam model PF09018[96], which corresponds to the tectivirus capsid protein. The recovered assemblies were annotated with prokka v1.14.6[87] using the PHROGs database[88]. To refine these annotations, our large collection of alphatectiviruses was used to build protein alignments for each protein in the PRD1 genome, using clustalo v1.2.4[85] and manually curated for quality. These alignments were then used to build hmm profile models with HMMER v3.3.1[97], to search them against the collected tectivirus MAGs. A representative selection of annotated MAGs was selected and visualized with clinker v0.0.28[91] and colored to show homology. Shaded connectors represent proteins with >0.3 sequence identity, while annotations with the same color represent significant ($p < 0.01$) homologs according to the HMMER search.

## Search for alphatectiviruses in metagenomic reads
Kraken2 v2.1.2[98] was used to search for the presence of alphatectiviruses reads in metagenomic datasets. A custom database was built by adding our new alphatectivirus assemblies to the default RefSeq viral reference library. With this database, a collection of reads from wastewater sequencing projects was searched. The SRA BioProject accession numbers of this collection can be found in Supplementary Dataset 2. The individual reads from each sequencing run that were classified as belonging to alphatectiviruses according to Kraken2 were extracted and mapped to the PRD1 reference genome with minimap2 v2.22. The resulting mapped reads were processed with samtools v1.6 and visualized with IGV v2.11.4[99].

## Phylogenetic trees
For the alphatectivirus tree, all previously published genomes and those collected in this study were aligned with clustalo v1.2.4. The resulting multiple sequence alignment was manually curated to ensure quality of the alignment. The tree was then built with iqtree v2.2.0.3[100], and visualized with iTOL v6.7[101].

For the *Fiersviridae* and *Inovirus* trees, the protein sequence of the RNA-dependent RNA polymerase (replicase) or the whole nucleotide content, respectively, were aligned. For the FtMidnight distal-tail protein trees, one alignment per protein was generated. All alignments were performed with clustalo v1.2.4, the trees were then generated with phyml v3.2.0[102] and visualized with FigTree v.1.4.4[103].

For the tectivirus ATPase tree, the amino acid sequences for protein P9 (ATPase) from all known tectiviruses were aligned with clustalo v1.2.4. This alignment was used to create an hmm profile model with HMMER, which was then used to search the amino acid sequences extracted from the annotated MAGs (see "Search for tectiviruses in metagenomic assembled genomes"). Significant hits were extracted and aligned to the model with HMMER. We also included in this alignment the previously metagenomic-assembled tectiviruses listed in Yutin et al.[50], and a selection of characterized representatives of the five tectivirus genera. A tree of the resulting ATPase alignment was built with phyml v3.2.0, and visualized with iTOL v6.7.

Specific alignment and tree building parameters can be found in the code repository. All accession numbers of sequences used to build these trees are listed in Supplementary Dataset 2.

## Electron microscopy
Carbon grids were glow discharged using a EMS100x Glow Discharge Unit for 30 s at 25 mA. High titer phage stocks were diluted 1:10 in water and 5 μL was adsorbed to the glow discharged carbon grid for 1 min. Excess sample was blotted with filter paper and the grids were washed once with water before staining with 1% uranyl acetate for 20 s. Excess stain was blotted with filter paper and the grids were air dried prior to examination with a Tecnai G2 Spirit BioTWIN Transmission Electron Microscope at the Harvard Medical School Electron Microscopy Facility.

## Reporting summary
Further information on research design is available in the Nature Portfolio Reporting Summary linked to this article.

## Data availability
Raw sequencing reads have been deposited in the NCBI BioProject database under accession number PRJNA954020. Accession numbers for novel phage genomes generated in this study can be found in Supplementary Dataset 1c. Raw data used in figures are available in a Github repository.

## Code availability
All code is available in a Github repository: https://github.com/baymlab/2023_QuinonesOlvera-Owen.

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

## Acknowledgements

We are grateful for the gifts of bacterial strains, plasmids, phages, or wastewater from the labs of Uli Klümper, Catherine Putonti, George O'Toole, Karine Gibbs, Jay Hinton, Pamela Silver, and Ameet Pinto. We thank the other instructors and students of the HMS Phages 2022 summer course: Thomas Bernhardt, Amelia McKitterick, Kate Hummels, Thomas Bartlett, Nawonh Charles, Melanie Justice, Tosin Bademosi, and Ahadu Molla, which was partially supported by the HHMI Science Education Alliance. N.Q.O. thanks the Marine Biological Laboratory at Woods Hole and all instructors from the 2019 Microbial Diversity course. Electron microscopy imaging and consultation were performed in the HMS Electron Microscopy Facility. Custom instrumentation was built with assistance from the Research Instrumentation core at Harvard Medical School. Computational work used the O2 cluster supported by the Research Computing Group at Harvard Medical School. This work was supported by the NIGMS of the National Institutes of Health (R35GM133700), the David and Lucile Packard Foundation, the Pew Charitable Trusts, Alfred P. Sloan Foundation and NSF grant IOS-2331228. N.Q.O. acknowledges support from Consejo Nacional de Ciencia y Tecnología (CONACYT, México). M.G.M., E.A.R., R.P., and J.S.P. acknowledge support from the Systems, Synthetic, and Quantitative Biology PhD program training award (T32GM135014). A.C.F. was supported in part by the NSF-Simons Center for Mathematical and Statistical Analysis of Biology at Harvard (award number #1764269), and the Harvard Quantitative Biology Initiative.

## Author contributions

N.Q.O., S.V.O., and M.B. conceived and designed the study. N.Q.O., S.V.O., L.M.M., A.C.F., O.J.M.D., K.P., C.E.S.C., R.P., and J.S.P. conducted experiments and acquired data. M.G.M. and E.A.R. contributed resources and data interpretation. N.Q.O., S.V.O., and M.B. analyzed the data and wrote the manuscript. All authors read and approved the manuscript.

## Competing interests

The authors declare no competing interests.

## Additional information

¹Department of Biomedical Informatics, Harvard Medical School, Boston, MA 02115, USA. ²Laboratory of Systems Pharmacology, Harvard Medical School, Boston, MA 02115, USA. ³Department of Microbiology, Harvard Medical School, Boston, MA 02115, USA. ⁴Boston University, Boston, MA 02215, USA. ⁵Department of Systems Biology, Harvard Medical School, Boston, MA 02115, USA. ⁶Roxbury Community College, Boston, MA 02120, USA. ⁷Broad Institute of MIT and Harvard, Cambridge, MA 02142, USA. ⁸These authors contributed equally: Natalia Quinones-Olvera, Siân V. Owen. ✉e-mail: sianvictoriaowen@gmail.com; baym@hms.harvard.edu

