## [Peer Review File · Nature Communications]

Diverse and abundant phages exploit conjugative plasmidsEditorial Note: This manuscript has been previously reviewed at another journal that is not operating a transparent peer review scheme. This document only contains reviewer comments and rebuttal letters for versions considered at *Nature Communications*.

Reviewer #1 (Remarks to the Author):

The authors have responded well to my previous concerns. This is a very impressive study that introduces an elegant isolation method. I only have very minor comments on this version.

L.101: maybe also give number of new genera?

Fig. 2: It is somewhat odd to talk about "unappreciated" diversity.

L119: I would strike the "greatly" from this sentence.

L330: perhaps give information on sequencing depth.

L344: I do not share this conclusion that there is a discordance since the metagenomic results should reflect the relative abundance in the total virome while the isolation results are a fairly narrow comparison in absolute abundance.

Reviewer #2 (Remarks to the Author):

The manuscript's transfer to Nature Communications is deemed highly appropriate for this study. Furthermore, the manuscript has been enhanced with additional data concerning specific aspects of the research, such as the FtMidnight phage and its host range.

The authors have also refined their language and interpretations to align more closely with the presented data, resulting in a significantly improved narrative.

The inclusion of new data on holins, as mentioned in the rebuttal letter, begins to elucidate the mechanisms behind host specificity. This is a valuable contribution to the manuscript. It is anticipated that future research by the authors or others will elucidate why certain phages exhibit a narrow host range due to their holin genes (e.g., PRDcerulean), whereas others demonstrate a broad host compatibility with their holin genes, affecting nearly all tested host strains (as illustrated in Figure 4c).

Overall, the authors have excelled in revising the manuscript and selecting a journal that better suits this study. Coupled with recent publications on conjugation plasmid-dependent phages (e.g., work by He et al., 2022, in *Water Research* and by Parra et al., 2023, in *Microbiology Spectrum* [both cited within this manuscript]), this work is likely to make a significant impact in the field. The innovative double-color method introduced here is particularly noteworthy and is expected to be widely adopted.

Minor Points:

- Line 174: The assertion that "the FtMidnight-like group of phages is the third example of a phage group that can use either contractile pilus structure fairly indiscriminately" may be considered overly assertive without comprehensive experimental support.
- Line 296ff: The methodology for assessing IncP plasmid-dependency, in contrast to species-specificity, is somewhat ambiguous, particularly regarding whether these assessments were conducted against multiple hosts. Given the preceding discussion that some phages utilize the conjugative pilus but still require specific adaptation to new hosts based on their holin characteristics, it seems that IncP-dependent phages could be categorized as either truly broad host or species-specific. Clarification or further explanation in the text might resolve this ambiguity or correct any misinterpretation.

Reviewer #3 (Remarks to the Author):

The authors thoroughly revised their manuscript, according to the comments of all three reviewers, which made the study much more compelling. The new data on the plasmid-dependent tailed phage and experiments showing the role of a holin as a host range determinant are particularly interesting. In my opinion, the manuscript can be published as is. Note that all family names should be written in italics (as per ICTV rules). There are a few places where they are not (e.g., "...Fiersviridae and the ssDNA filamentous Inoviridae..."). Please check and correct (this can be done when preparing the final documents).

REVIEWERS' COMMENTS

Reviewer #1 (Remarks to the Author):

The authors have responded well to my previous concerns. This is a very impressive study that introduces an elegant isolation method. I only have very minor comments on this version.

L.101: maybe also give number of new genera?

We have added this at line 100-101

Fig. 2: It is somewhat odd to talk about “unappreciated” diversity.

“Unappreciated” has been replaced with “expanded”

L119: I would strike the “greatly” from this sentence.

“greatly” has been removed

L330: perhaps give information on sequencing depth.

We have added this information in the results section at line 280

L344: I do not share this conclusion that there is a discordance since the metagenomic results should reflect the relative abundance in the total virome while the isolation results are a fairly narrow comparison in absolute abundance.

We agree with the reviewer that the comparison between metagenomic and isolation based abundance quantification is difficult, and we believe we have provided a nuanced interpretation and discussion of these results

Reviewer #2 (Remarks to the Author):

The manuscript's transfer to Nature Communications is deemed highly appropriate for this study. Furthermore, the manuscript has been enhanced with additional data concerning specific aspects of the research, such as the FtMidnight phage and its host range.

The authors have also refined their language and interpretations to align more closely with the presented data, resulting in a significantly improved narrative.

The inclusion of new data on holins, as mentioned in the rebuttal letter, begins to elucidate the mechanisms behind host specificity. This is a valuable contribution to the manuscript. It is anticipated that future research by the authors or others will elucidate why certain phages exhibit a narrow host range due to their holin genes (e.g., PRDcerulean), whereas others demonstrate a broad host compatibility with their holin genes, affecting nearly all tested host strains (as illustrated in Figure 4c).

Overall, the authors have excelled in revising the manuscript and selecting a journal that better suits this study. Coupled with recent publications on conjugation plasmid-dependent phages (e.g., work by He et al., 2022, in Water Research and by Parra et al., 2023, in Microbiology Spectrum [both cited within this manuscript]), this work is likely to make a significant impact in the field. The innovative double-color method introduced here is particularly noteworthy and is expected to be widely adopted.

Minor Points:

- Line 174: The assertion that "the FtMidnight-like group of phages is the third example of a phage group that can use either contractile pilus structure fairly indiscriminately" may be considered overly assertive without comprehensive experimental support.

We modified the wording of this sentence to “ the FtMidnight-like group of phages is the third example of a phage group that can adapt to use either contractile pilus structure in short evolutionary timescales”. We did not intend to imply these phages could use either pilus, and the modified sentence is more accurately supported by our data.

- Line 296ff: The methodology for assessing IncP plasmid-dependency, in contrast to species-specificity, is somewhat ambiguous, particularly regarding whether these assessments were conducted against multiple hosts. Given the preceding discussion that some phages utilize the

conjugative pilus but still require specific adaptation to new hosts based on their holin characteristics, it seems that IncP-dependent phages could be categorized as either truly broad host or species-specific. Clarification or further explanation in the text might resolve this ambiguity or correct any misinterpretation.

The reviewer is correct that our quantification of “plasmid-depedendent phages” in Figure 6A is limited by the inability to detect plasmid-dependent phages with host specificities not represented in our assay. Unfortunately we do not have a culture-based way to overcome this limitation and the data must be interpreted with these limitations in mind. We discuss this at line 261.

Reviewer #3 (Remarks to the Author):

The authors thoroughly revised their manuscript, according to the comments of all three reviewers, which made the study much more compelling. The new data on the plasmid-dependent tailed phage and experiments showing the role of a holin as a host range determinant are particularly interesting. In my opinion, the manuscript can be published as is. Note that all family names should be written in italics (as per ICTV rules). There are a few places where they are not (e.g., “...Fiersviridae and the ssDNA filamentous Inoviridae...”). Please check and correct (this can be done when preparing the final documents).

We have fixed all instances of non-italicised family names in the revised version.